# STOCHASTIC GRADIENT LANGEVIN DYNAMICS THAT EXPLOIT NEURAL NETWORK STRUCTURE

**Zachary Nado, Jasper Snoek**
Google Brain
znado@google.com

**Bowen Xu, Roger Grosse,**
**David Duvenaud**
University of Toronto
Vector Institute

**James Martens**
DeepMind

## ABSTRACT

Tractable approximate Bayesian inference for deep neural networks remains challenging. Stochastic Gradient Langevin Dynamics (SGLD) offers a tractable approximation to the gold standard of Hamiltonian Monte Carlo. We improve on existing methods for SGLD by incorporating a recently-developed tractable approximation of the Fisher information, known as K-FAC, as a preconditioner.

## 1 INTRODUCTION

Accurately tacking model uncertainty in deep neural networks is important for tasks such as decision making, planning and exploration in reinforcement learning, and medical diagnosis. However, tractable approximate Bayesian inference in neural networks remains challenging. Markov chain Monte Carlo (MCMC) methods are appealing, as they allow us to draw samples from the approximate posterior without having to compute and store sufficient statistics of the distribution over parameters.

A variety of methods have refined traditional MCMC methods by reshaping the sampling distribution using a Riemannian tensor such as the Fisher information (Girolami & Calderhead, 2011; Patterson & Teh, 2013; Ahn et al., 2012; Ma et al., 2015; Mandt et al., 2016). Unfortunately, the full Fisher matrix is rarely tractable, and the diagonal approximation is usually poor.

We build an approximate Riemannian tensor using recent advances in approximating the Fisher information for deep neural networks (Martens & Grosse, 2015) and show that it improves sampling in Stochastic Gradient Langevin Dynamics.

## 2 STOCHASTIC GRADIENT MCMC

Neal (1994) introduced the gold standard method for modeling uncertainty in deep networks, Hamiltonian Monte Carlo (HMC), bringing to bear methods from Monte Carlo simulation for lattice field theory (Duane et al., 1987). Unfortunately, HMC remains intractable for large datasets, since it requires full-batch computations and thus does not lend itself to stochastic updates which depend only on mini-batches. Welling & Teh (2011) introduced Stochastic Gradient Langevin Dynamics (SGLD) as a stochastic mini-batch approximation to HMC. SGLD is appealing as it results in a simple modification to standard Robbins-Monro stochastic gradients where standard Gaussian noise, scaled by the learning rate $\epsilon$, is added to the gradient updates of the parameters $\theta$ at each time step $t$ as follows:

$$\theta_{t+1} = \theta_t - \frac{\epsilon}{2}\left(\nabla \log p(\theta_t) + \frac{n}{N}\sum_{i=1}^{N}\nabla \log p(\mathbf{x}_i|\theta_t))\right) + \sqrt{\epsilon}\nu, \quad \nu \sim \mathcal{N}(0, I) \tag{1}$$

Unfortunately, Langevin Dynamics, which SGLD is approximating, suffers from poor mixing and highly correlated samples. A variety of methods have built upon that work to mitigate these issues. Patterson & Teh (2013) proposed Stochastic Gradient Riemannian Langevin Dynamics (Reimannian SGLD), which uses a positive-definite preconditioning matrix to incorporate the geometry of the log-probability function into the shape of the sampling distribution. Writing $g()$ for the gradient of the likelihood times prior for brevity, Riemannian SGLD uses updated that depend on a positive-definite matrix $\mathbf{H}$:

$$\theta_{t+1} = \theta_t - \epsilon\,\mathbf{H}^{-1}\,g(\theta_t) + \sqrt{\epsilon}\,\mathbf{H}^{-\frac{1}{2}}\,\nu, \qquad \nu \sim \mathcal{N}(0, I) \tag{2}$$

Ahn et al. (2012) introduced Stochastic Gradient Fisher Scoring, which scales the gradient updates and noise using the empirical Fisher information matrix. Chen et al. (2014) proposed a stochastic gradient HMC algorithm. Li et al. (2016) used the popular RMSProp algorithm to form a diagonal preconditioning matrix and show that it outperforms standard SGLD. Ma et al. (2015) provide a nice summary of stochastic gradient MCMC algorithms and also propose a Riemannian version. While these methods propose the use of a Riemannian tensor, they rely on either computing the full Fisher information or a diagonal approximation. As such, they could all benefit from an accurate and tractable approximation to the Fisher. (Marceau-Caron & Ollivier, 2017) recently proposed a similar approach using an alternative "quasi-diagonal" approximation to the Fisher information (Ollivier, 2015).

## 3 KRONECKER FACTORED APPROXIMATE CURVATURE

Natural gradient descent preconditions the gradient according to the Fisher information matrix $\mathbf{F} = \mathrm{Cov}(\nabla_\theta \log p(y|x))$, i.e. $\tilde{\nabla}_\theta h = \mathbf{F}^{-1} \nabla_\theta h$. For many neural network models, this can be interpreted as generalized Gauss-Newton, a second-order algorithm Martens (2014). But just as with the Hessian, the dimension of $\mathbf{F}$ is the number of parameters of the model, so computing or storing it is infeasible.

Kronecker-Factored Approximate Curvature (K-FAC) (Martens & Grosse, 2015) is a tractable approximation to $\mathbf{F}$ based on a probabilistic approximation to the backprop computations. In particular, if we make the approximation that the weight derivatives in different layers are independent, and that the activations are independent of the backpropagated derivatives, then the approximate Fisher matrix $\hat{\mathbf{F}}$ has a block diagonal structure with a block $\hat{\mathbf{F}}_\ell$ for each layer $\ell$ of the network. Each of these blocks decomposes as a Kronecker product

$$\hat{\mathbf{F}}_\ell = \mathbf{A}_{\ell-1} \otimes \mathbf{S}_\ell,$$

where $\mathbf{A} = \mathbb{E}[\mathbf{a}_\ell \mathbf{a}_\ell^\top]$ is the uncentered covariance of the activations, and $\mathbf{S} = \mathrm{Cov}(\mathcal{D}\mathbf{s}_\ell)$ is the covariance of the log-likelihood derivatives $\mathcal{D}\mathbf{s}_\ell = \nabla_{\mathbf{s}_\ell} \log p(y|x)$ with respect to the pre-activations $\mathbf{s}$. This allows the natural gradient to be computed efficiently:

$$\hat{\mathbf{F}}^{-1} \nabla_\theta h = \begin{pmatrix} \mathrm{vec}\left(\mathbf{S}_1^{-1}(\nabla_{\mathbf{W}_1} h)\mathbf{A}_0^{-1}\right) \\ \vdots \\ \mathrm{vec}\left(\mathbf{S}_L^{-1}(\nabla_{\mathbf{W}_L} h)\mathbf{A}_{L-1}^{-1}\right) \end{pmatrix} \tag{3}$$

Due to the mixed-product property of the Kronecker product, the Cholesky factorization of a Kronecker product of two matrices is equal to the Kronecker product of the individual Cholesky factorizations. Thus we can use this property to efficiently sample using the noise covariance $\mathbf{H}^{-\frac{1}{2}}$.

## 4 EMPIRICAL EVALUATION

### 4.1 SIMULATED DATA

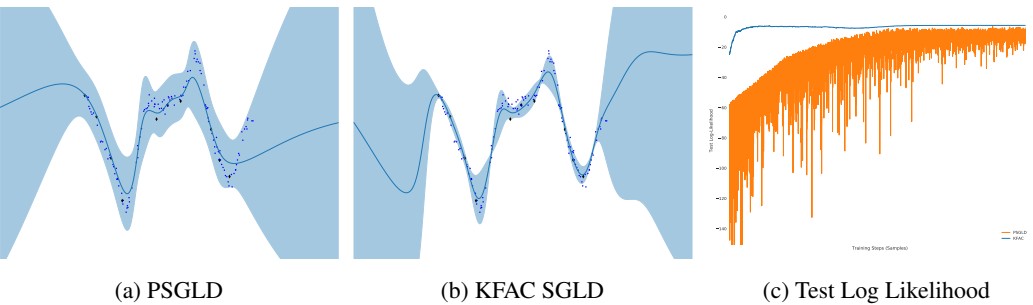

(a) PSGLD       (b) KFAC SGLD       (c) Test Log Likelihood

Figure 1: Qualitative plots and test log-likelihood of the sample distribution of our method compared to preconditioned SGLD (Li et al., 2016).

In this section we apply our method to a simple one-dimension function sampled from a Gaussian process with squared exponential covariance function. In Figures 1a and 1b we plot the sample distribution using preconditioned SGLD (Li et al., 2016) and our method respectively. In Figure 1c we plot the log-likelihood of the samples on test data as a function of training steps.

## 4.2 BOSTON HOUSING DATA

In this section we apply our method to the Boston housing dataset, which is a relatively small regression problem of 506 examples in 13 dimensions. We used 10 samples of our model taken with a thinning interval of every 100 iterations to compute the test values, with a mini-batch size of 32. Our model is a single hidden layer neural network of width 50 with ReLU activations. In Figure 2a we plot the effective sample size (ESS) of our method, calculated using the equation from (Homan & Gelman, 2014) (lower is better). In Figure 2b we plot the log-likelihood of the samples on test data as a function of training steps. We note that in addition to achieving a better ESS than with SGLD alone, with K-FAC preconditioning we are able to achieve a log-likelihood of $-2.226 \pm 0.069$, which is higher than the $-2.574 \pm 0.089$ achieved in (Hernández-Lobato & Adams, 2015).

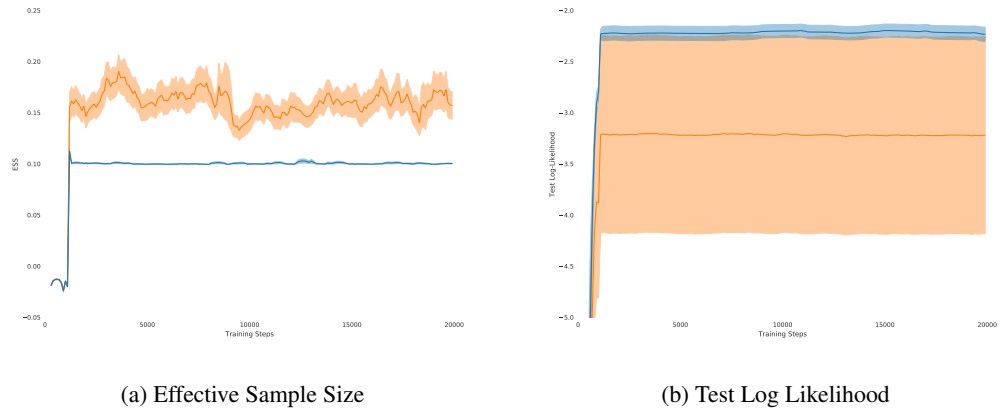

(a) Effective Sample Size          (b) Test Log Likelihood

Figure 2: Effective sample size and test log-likelihood on the Boston housing dataset of our method compared to preconditioned SGLD (Li et al., 2016). The plots include a burn in period of 300 steps and thinning each 100 samples.

## 5 CONCLUSION

In this paper we develop a method for scalable approximate Bayesian inference for deep neural networks that is both efficient and effective by preconditioning SGLD with a tractable approximation of the Fisher information. Initial results of this approach are quite promising. By using the Kronecker-Factored Approximate Curvature approximation of the Fisher information as a preconditioner, we are able to achieve better log-likelihood performance on the Boston housing data set than existing approximate Bayesian inference methods. We think this is a promising avenue of further research and wish to continue this work by applying this technique to larger scale classification and reinforcement learning problems. In addition, it would be interesting to develop the application of these techniques to Hamiltonian Monte Carlo and other Langevin methods.

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
