# OpenReview forum: "STOCHASTIC GRADIENT LANGEVIN DYNAMICS THAT EXPLOIT NEURAL NETWORK  STRUCTURE"
_ICLR.cc/2018/Workshop — Accept_

### Official Review · AnonReviewer3 · 2018-03-09
**Useful fast sampling method**

**Rating:** 8
**Confidence:** 5

**Review:**

This paper provides an effective fisher information approximation method K-FAC to SGLD, the experiments show that the sampler out-performs pre-conditioned version of SGLD.

Comments:
The author mentioned cited Riemannian based samplers, which can resort to the diagonal approximate version that author claimed to be inaccurate. I would expect to see a comparison between the diagonal approximate Riemannian vs K-FAC based sampler.

---

### Official Review · AnonReviewer1 · 2018-03-10
**Promising early result**

**Rating:** 7
**Confidence:** 4

**Review:**

The authors propose to use the Kronecker-Factored Approximate Curvature (K-FAC) approximation of the Fisher information as a preconditioner of SGLD. The experiment result shows that it's a promising direction.

---

### Official Review · AnonReviewer2 · 2018-03-10
**Application of K-Fac formulation to Fisher matrix improves conditioned SGLD**

**Rating:** 6
**Confidence:** 4

**Review:**

The authors provide an application of the Kronecker Factored curvature proposed in Martens et. al. (ICML'15) to preconditioned SGLD, allowing one to efficiently approximate the product of the Fisher metric with the standard stochastic gradient (known as the "natural gradient"). The empirical evaluation shows some improvement over pSGLD in terms of test log likelihood.

Overall the paper does not have any theoretical contributions over the ICML'15 work referenced above, and experimental results comparing against the diagonal Riemannian approximations (SGRLD) with a runtime comparison would have been nice.

---

### Decision · Program_Chairs · 2018-03-20
**ICLR 2018 Workshop Acceptance Decision**

**Decision:**

Accept

**Comment:**

Congratulations, your paper was accepted to the ICLR workshop.